# A Method for Selection of Coffee Varieties Resistant to *Fusarium stilboides*

**Getrude Okutoyi Alworah [1], Joshua Ondura Ogendo [2], Joseph Juma Mafurah [2], Elijah Kathurima Gichuru [1],\*, Douglas Watuku Miano [3] and Otieno Oliver Okumu [3]**

[1] Department of Plant Pathology, KALRO-Coffee Research Institute, P.O. Box 4, Ruiru 00232, Kenya; getrude.alworah@kalro.org
[2] Department of Crops, Horticulture and Soils, Egerton University, P.O. Box 536, Egerton 20115, Kenya; jogendo@egerton.ac.ke (J.O.O.); joseph.mafurah@egerton.ac.ke (J.J.M.)
[3] Department of Plant Science and Crop Protection, University of Nairobi, P.O. Box 29053, Kangemi 00625, Kenya; dmiano@uonbi.ac.ke (D.W.M.); oliverotieno182@gmail.com (O.O.O.)
\* Correspondence: ekgichuru@gmail.com or elijah.gichuru@kalro.org

**Abstract:** Fusarium bark disease (FBD) of coffee caused by *Fusarium stilboides* Steyaert has been associated with yield losses and tree death in coffee-growing countries, including Kenya. Varietal resistance is of utmost importance in managing the disease, and the continued increase in host resistance is considered sufficient to control the disease. Therefore, this study investigated the response of four coffee cultivars to *F. stilboides*. Fifteen hypocotyls from *Coffea arabica* (Ruiru 11, Batian and SL 28), plus two other coffee species, *Coffea canephora* (Robusta coffee) and *Coffea liberica* (Liberica coffee), were inoculated with various concentrations of three *F. stilboides* isolates (NRF 930/18, TN002B (I), BS008B (A)) using both the injection and drenching methods. The stem injection method was performed by injecting one microliter of the *F. stilboides* spore solution into the hypocotyl. In the drenching method, coffee seedlings had their roots cut and transplanted in a transplanting mixture, after which 10 mL of the *F. stilboides* spore solution was applied to the soil surface in each pot. The inoculated hypocotyls were incubated at 24 ± 2 °C for 105 days. The plants were watered regularly as necessary while the development of symptoms was observed and assessed weekly. Disease severity was evaluated using a modified scale of 0–4, while incidence was scored as a percentage of infected seedlings. Coffee seedlings inoculated with *F. stilboides* developed symptoms such as wilting, stunted growth and defoliation. In the first season, the coffee cultivars of Ruiru 11 and Liberica inoculated with *Fusarium stilboides* had pronounced severity compared to other cultivars. In the second season, the coffee cultivars Batian and Ruiru 11 had pronounced severity compared to other cultivars. The isolate TN002B (I) was observed to be highly virulent as compared to other isolates. The variation in response to disease infection exhibited by the four cultivars presents a key input in breeding programs for resistance to *F. stilboides*.

**Keywords:** Fusarium-bark disease; inoculation; resistance; pathogenicity; severity

## 1. Introduction

Coffee production in Kenya is concentrated in the Western, Rift Valley and Central regions in the high potential areas between 1400 and 2200 m above sea level (masl) with temperatures ranging between 15 and 24 °C and red, deep, and well-drained volcanic soils [1]. Worldwide, the coffee tree is affected by several fungal diseases from the genera *Colletotrichum*, *Hemileia* [2] and *Fusarium* [3], causing yield losses directly or indirectly. Coffee bark disease, also known as "*Fusarium* bark disease", is a serious and destructive disease of coffee. The disease is caused by the soil-borne fungus, *Fusarium stilboides*, the asexual stage (anamorph), the sexual stage (teleomorph) being *Gibberella stilboides* [4,5]. The disease mainly infects coffee plants through wounds, especially those caused during pruning and mechanical weeding. It requires humid conditions and an optimum temperature of 30 °C

in the field and 25 °C in culture for its spores to germinate without free water [6,7]. The disease was first detected in Tanzania in 1932 and later in Malawi, where it caused serious economic losses and was largely controlled by selecting resistant cultivars. The disease has since been reported in Burundi, Ethiopia, Kenya, Madagascar, Rwanda, Tanzania and Uganda. In Kenya, the disease has become a major threat to coffee production and has been reported in the major coffee growing zones. Coffee bark disease was regarded as a minor disease in Kenya as it was never reported to be causing any significant yield losses until 2013, when it was reported to cause havoc in Nyeri County and progressively became more severe in higher altitudes. Preliminary studies showed that the disease caused up to 100% crop losses and was wiping out coffee farms [8]. There are minimal studies on managing the disease, with farmers mainly uprooting and replacing the infected trees [5,8]. This practice is, however, costly and the newly planted seedlings also get infected, resulting in a cycle of planting -uprooting and planting again. Selecting and developing resistant cultivars would therefore offer a sustainable and effective disease management strategy. The objective of this study was therefore to develop a suitable method for identifying coffee bark disease-resistant varieties that would offer effective management of the disease.

## 2. Materials and Methods

### 2.1. Plant Materials and Planting of Coffee Seeds

Three thousand coffee seeds of three commercial cultivars of *Coffea arabica*—Ruiru 11, Batian and SL 28 which have a narrow genetic base with a diversity index of 10% compared to Timor hybrid (HDT) series, plus two other coffee species; *Coffea canephora* (Robusta) and *Coffea liberica* (Liberica coffee), were dehusked and planted in coarse sand in 1.2 L lunch boxes. The seeds were monitored and watered daily for 12 weeks until 90% germination was achieved. The seedlings were retained in the sand for 2 weeks until the two cotyledons unfolded. Coffee seedlings were used despite their weak resistance because they can still show a level of resistance that may discriminate between genotypes. This early detection of resistance can guide breeding efforts and help select promising varieties for further evaluation.

### 2.2. Preparation, Purification, and Inoculation of Fungal Isolates

Confirmed isolates of *F. stilboides* that were preserved at the Plant Pathology Laboratory at KALRO-Coffee Research Institute (KALRO-CRI), Ruiru, Kenya, were used in the study. The fungal isolates were isolated from symptomatic coffee tree samples and plated in petri dishes containing Potato Dextrose Agar (PDA) culture media. After ten days of growth, the cultured isolates were purified by the single conidia method and stored in Synthetischer Nährstoffarmer agar (SNA) culture media for use in inoculum preparation [9,10]. The treatments were three *F. stilboides* isolates from Kirinyaga (NRF 930/18), Tharaka Nithi (TN002B) and Busia (BS008B) counties. Inoculation was performed on nine weeks old seedlings of three commercial cultivars of *Coffea arabica* (Ruiru 11, Batian and SL 28), plus two other coffee species, *Coffea canephora* (Robusta) and *Coffea liberica* (Liberica coffee). The experiment was performed in two seasons: October 2019 to September 2020 and October 2020 to September 2021.

### 2.3. Harvesting and Quantification of Inoculum

Six-day old *F. stilboides* cultures on PDA media were cut using a sterile scalpel and transferred to a 500 mL conical flask. Fifteenpieces of *F. stilboides* culture were transferred in one conical flask, and 150 mL of sterile distilled water was added. The flasks were then transferred to a wrist action flask shaker. The flasks were shaken at a speed of 80 rounds per minute for 30 min before harvesting the spores [11]. The spores were harvested into a 500 mL conical flask by passing them through two-layered sterile muslin cloth and then through a funnel into the conical flask. After harvesting the spores, a drop was transferred to a hemocytometer slide using a teat pipette and observed under a compound microscope [12]. The total number of spores on the two sides was used to determine the

number of spores in 1 mL of the harvest, and the concentration was adjusted to $10^5$ and $10^6$ spores per mL.

### 2.4. Formulation of Potting Mixtures, Transplanting and Pathogen Inoculation

Transplanting potting mixture was formulated in a ratio of 3:2:1 by mixing topsoil (18 Kg), sand (12 Kg), and manure (6 Kg). Then 100 g of Triple Super Phosphate (TSP) fertilizer was added and thoroughly mixed. The topsoil (0–20 cm depth) was retrieved from part of the farm that had been left fallow. Portions of the mixture were put into 1.2-L plastic lunch boxes (Homeware® Food containers, KenPoly manufacturers Limited, Kenya). After over 90% of the seedlings had fully opened cotyledons, fifteen seedlings from each variety were inoculated and transplanted into the lunch boxes with the transplanting mixture.

Two inoculation methods were tested for effectiveness: stem injection and soil drench. The stem injection method was performed as was indicated by Li et al. [13] by creating a hole on the stem using a sterile disposable needle after which one microliter of the *F. stilboides* spore solution was injected into the hypocotyl of the seedling at about the $\frac{3}{4}$ point above the soil level. For the drenching method, coffee seedlings had their roots cut and transplanted in the transplanting mixture after which 10 mL of the *F. stilboides* spore solution was applied to the surface of the soil [11]. Seedlings in the control plots were injected with sterile distilled water. The inoculated seedlings were maintained at room temperature for 2 weeks, then moved to a greenhouse and laid out in a complete block design. Watering was performed daily while nutrition, pest control and other agronomic practices were performed on an as-needed basis up to the 15th week when the disease data were taken.

### 2.5. Confirmation of the Pathogen

*Fusarium stilboides* was re-isolated from infected coffee seedlings. The seedlings were uprooted, washed with sterile distilled water to remove the soil, and then sectioned longitudinally from the point of inoculation. The pieces were rinsed in distilled water and surface sterilized by soaking in 2% sodium hypochlorite solution for 1 min, followed by a dip in 70% ethanol for 1 min, rinsed in sterile distilled water four times and blot-dried using paper towels. Five pieces were placed in Potato Dextrose Agar (PDA) and amended with antibiotics. The plates were incubated at $24 \pm 2\ ^\circ C$, and the cultures were scored for the presence of *F. stilboides*.

### 2.6. Data Collection and Analysis

The datad on severity were based on the symptoms such as the wilting of leaves that became dark brown and brittle but remained attached to the twigs, stunted growth and defoliation. Disease severity was evaluated using a modified scale of 0–4 where 0—healthy plants, 1—<10% stem lesion, 2—(>10%–<30% stem colonization, 3—(>30%–<50% stem colonization), and 4—(>50% stem colonization) (Tables 1 and 2) whereas disease incidence was the percentage of the infected plants determined using the equation proposed by Cooke et al. [14]. The AreaUnder the Disease Progress Curve (AUDPC) was determined using a simple midpoint (trapezoidal) rule [15], which breaks up a disease progress curve into a series of trapezoids, calculating the area of each and then adding up the areas. The disease severity and incidence data for Fusarium bark disease were subjected to Analysis Of Variance (ANOVA) using SAS Version 9.3 software. Means were separated using Fisher's Least Significant Difference at $p < 0.05$.

**Table 1.** Coffee bark disease symptoms severity index for inoculation by drenching.

| Score | 0 (0%) | 1 (<25% Infection at Soil Level) | 2 (>25%–<50% Infections from Soil Level Upwards) | 3 (>50%–<75% Infection from Soil Level Upwards) | 4 (>75% Infection) |
|---|---|---|---|---|---|
| Description | Healthy plants, turgid and shiny stems and leaves. No visible lesion at the soil level. | Small visible brown water-soaked lesions at the soil level. Flaccid leaves and stem. **Note:** There may be wilting but no visible lesion at the soil level. | Dark brown/black lesions at the soil level extending upwards and wilting. **Note:** There may be extensive wilting and stem drooping even without visible lesions at the soil level upwards. | ExtendedExtened dark brown/black lesions from soil level upwards. Wilting, drying, and yellowing of leaves and stem drooping. | Dead, dried, and rotten plants. Extended dark brown/black lesions colonizing more than 75% of the stem area or the entire plant. |
| Visual aid |  |  |  |  |  |

**Table 2.** Coffee bark disease symptoms severity index for inoculation by injection.

| Score | 0 (0%) Stem Colonization | 1 (<10% Stem Colonization) | 2 (>10%–<30% Stem Colonization) | 3 (>30%–<50% Stem Colonization) | 4 (>50% Stem Colonization) |
|---|---|---|---|---|---|
| Description | Healthy plants/scab (healing injection point)-(no visible lesion even under magnifying lens). Turgid and shiny stems and leaves. | Turgid and shiny stems and leaves with tiny brown/black lesions at the injection point. Spores may be visible using a magnifying lens. **Note:** There may be no visible lesions at the point of injection, but the stems and Leaves are flaccid but not drooping. Check for lesions along the stem or at soil level. | Extended dark brown/black lesions at the injection point, either extending upwards or downwards. There may be extended lesions from the soil level upwards and wilted leaves. Spores may be visible using a magnifying lens. Drooping and/or drying stem above the lesion and wilted leaves. | Extended dark brown/black lesions at the injection point, either extending upwards or downwards. There may be extended lesions from the soil level upwards and wilted leaves. Spores may be visible using a magnifying lens. Drooping and/or drying stem above the lesion and wilted leaves. | Extended dark brown/black lesions at the injection point, either extending upwards or downwards. There may be extended lesions from soil level upwards and wilted leaves. Spores may be visible using a magnifying lens. Drooping and/or drying stem above the lesion and wilted leaves. |
| Visual aid |  |  |  |  |  |

## 3. Results

### 3.1. Pathogenicity of Isolates of Fusarium stilboides Isolated from Infected Coffee Trees

The analysis ofvariance indicated a significant difference ($p < 0.05$) in the severity of *F. stilboides* in the plots inoculated using the two methods (injection and drenching), among the two inoculum concentration levels, the varieties, and the interaction between the isolates. Data collected at the peak of disease progression (15th week) indicated strong interactions between the variety and inoculation methods (Table 3). Additionally, there was a significant difference ($p < 0.05$) in disease severity among the different varieties, inoculum concentrations, and isolates. Furthermore, the interaction between the pathogen isolates and the inoculum concentration was only significant in season one (Table 3).

**Table 3.** First and second season Analysis of Variance of the Severity of *Fusarium stilboides*.

| Season One | | | | | |
| --- | --- | --- | --- | --- | --- |
| **Source** | **df** | **SS** | **MS** | **F Value** | **Pr > F** |
| Inoculation method | 1 | 5029.45 | 5029.45 | 29.83 | <0.0001 * |
| Inoculum concentration | 3 | 15,550.01 | 5183.34 | 30.74 | <0.0001 * |
| Pathogen isolate | 2 | 458.79 | 229.39 | 1.15 | 0.3185 |
| Variety | 4 | 3320.34 | 830.09 | 4.16 | 0.0028 * |
| Variety × inoculation method | 4 | 713.07 | 178.27 | 1.06 | 0.3782 |
| Variety × inoculum concentration | 8 | 395.14 | 49.39 | 0.25 | 0.9811 |
| Variety × pathogen isolate | 8 | 1812.03 | 226.5 | 1.13 | 0.3403 |
| Inoculation method × inoculum concentration | 3 | 1500.23 | 500.08 | 2.97 | 0.0326 |
| Pathogen isolate × inoculum concentration | 4 | 1981.39 | 495.35 | 2.48 | 0.0444 * |
| Variety × inoculation method × inoculum concentration | 12 | 2150.79 | 179.23 | 1.06 | 0.392 |
| Variety × pathogen isolate × inoculum concentration | 16 | 3740.68 | 233.79 | 1.17 | 0.2915 |
| **Season Two** | | | | | |
| Inoculation method | 1 | 365.07 | 365.07 | 2.74 | 0.0664 |
| Inoculum concentration | 3 | 17,210.01 | 5736.67 | 24.01 | <0.0001 * |
| Pathogen isolate | 2 | 6013.16 | 3006.58 | 13.46 | <0.0001 * |
| Variety | 4 | 3849.08 | 962.27 | 4.31 | 0.0022 * |
| Variety × inoculation method | 4 | 2350.63 | 587.66 | 2.46 | 0.0461 * |
| Variety × inoculum concentration | 8 | 2504.41 | 313.05 | 1.4 | 0.1962 |
| Variety × pathogen isolate | 8 | 2277.21 | 284.65 | 1.27 | 0.2573 |
| Inoculation method × inoculum concentration | 3 | 265.37 | 88.46 | 0.37 | 0.7746 |
| Pathogen isolate × inoculum concentration | 4 | 800.36 | 200.09 | 0.9 | 0.4671 |
| Variety × inoculation method × inoculum concentration | 12 | 2040.14 | 170.01 | 0.71 | 0.7399 |
| Variety × pathogen isolate × inoculum concentration | 16 | 4358.16 | 272.39 | 1.22 | 0.2532 |

* $p < 0.05$.

### 3.2. Effect of Various Isolates of F. stilboides on Disease Severity and Progression

Disease severity was significantly higher in varieties inoculated with isolate 1 (NRF 930/18) than in the other isolates in both Season 1 and Season 2, regardless of the concentration, inoculation method, and variety (Figure 1). However, there werewrer no significant differences in the severity of Fusarium bark disease in seedlings inoculated with isolates 2 [N002B (I) and 3 [BS008B (A). In the first season (Figure 1b), the Area Under Disease Progress Curve (AUDPC) for isolate 1 (NRF 930/18) and isolate 3 [BS008B (A] was significantly higher than the other isolates and the control. However, in the second season (Figure 1c), the AUDPC for isolate 1 (NRF 930/18) was significantly higher compared to other isolates. It is also important to note that a clear distinction in terms of AUDPC was observed immediately after the 7th week of observation for both seasons.

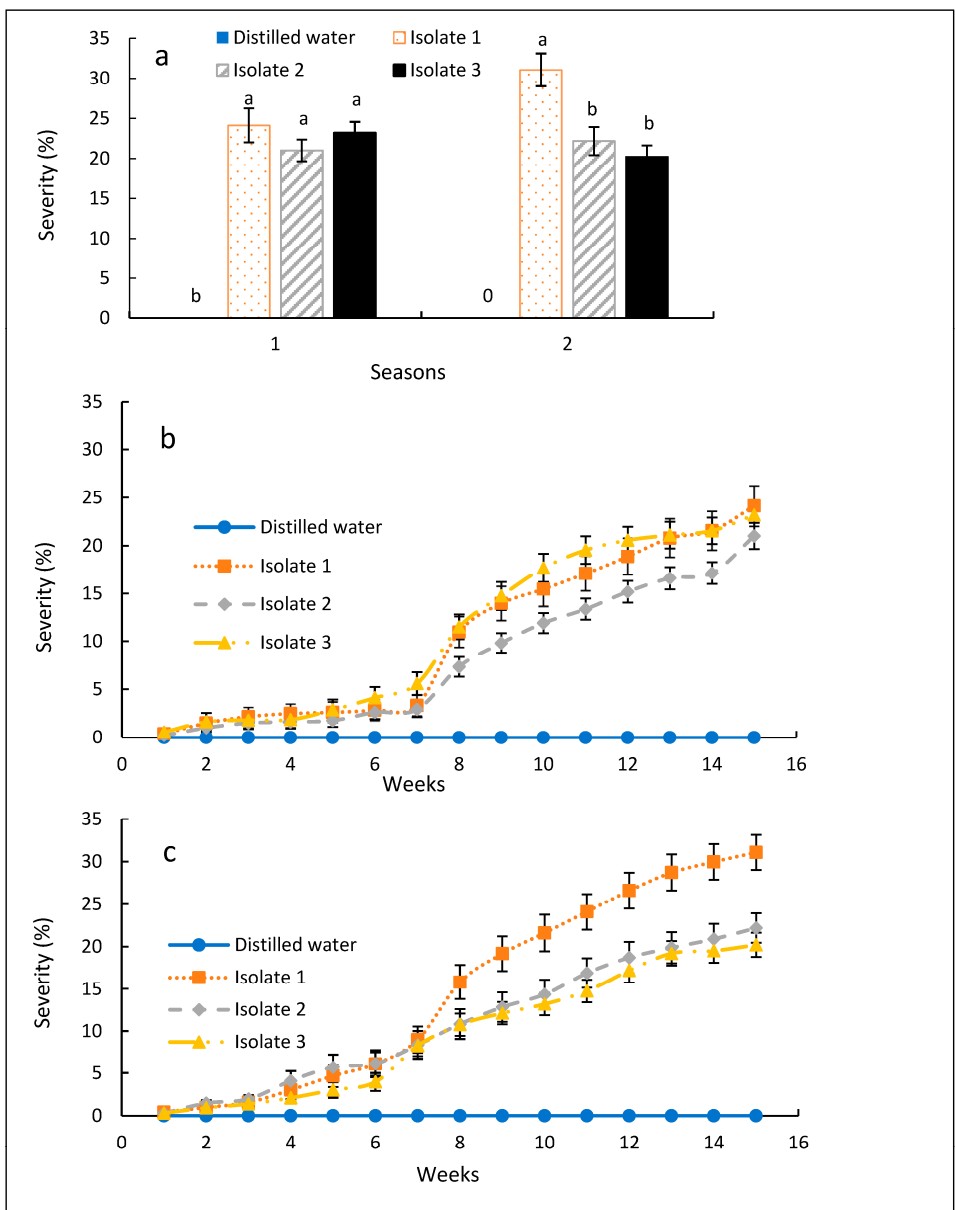

**Figure 1.** Severity of Fusarium bark disease on hypocotyls inoculated with different isolates (**a**) at Week 15, (**b**) disease progression for Season 1 and (**c**) Season 22. Bars with the same letters are not significantly different at $p < 0.05$.

### 3.3. Severity of Fusarium Bark Disease of Coffee

In the first season, the severity of the Fusarium bark disease for variety Ruiru 11 (*C. arabica*) was significantly higher ($p < 0.05$) when compared to other varieties whereas the lowest disease severity was observed in seedlings of Robusta. The same trend was observed in the second season; however, higher disease severity was reported in Liberica coffee seedlings (Figure 2). In terms of the inoculation method, seedlings injected with FBD accumulated more disease than those inoculated through drenching. Generally, in both seasons, Ruiru 11 accumulated more disease when compared to the other varieties, while, Robusta constantly accumulated less disease (Figures 3 and 4).

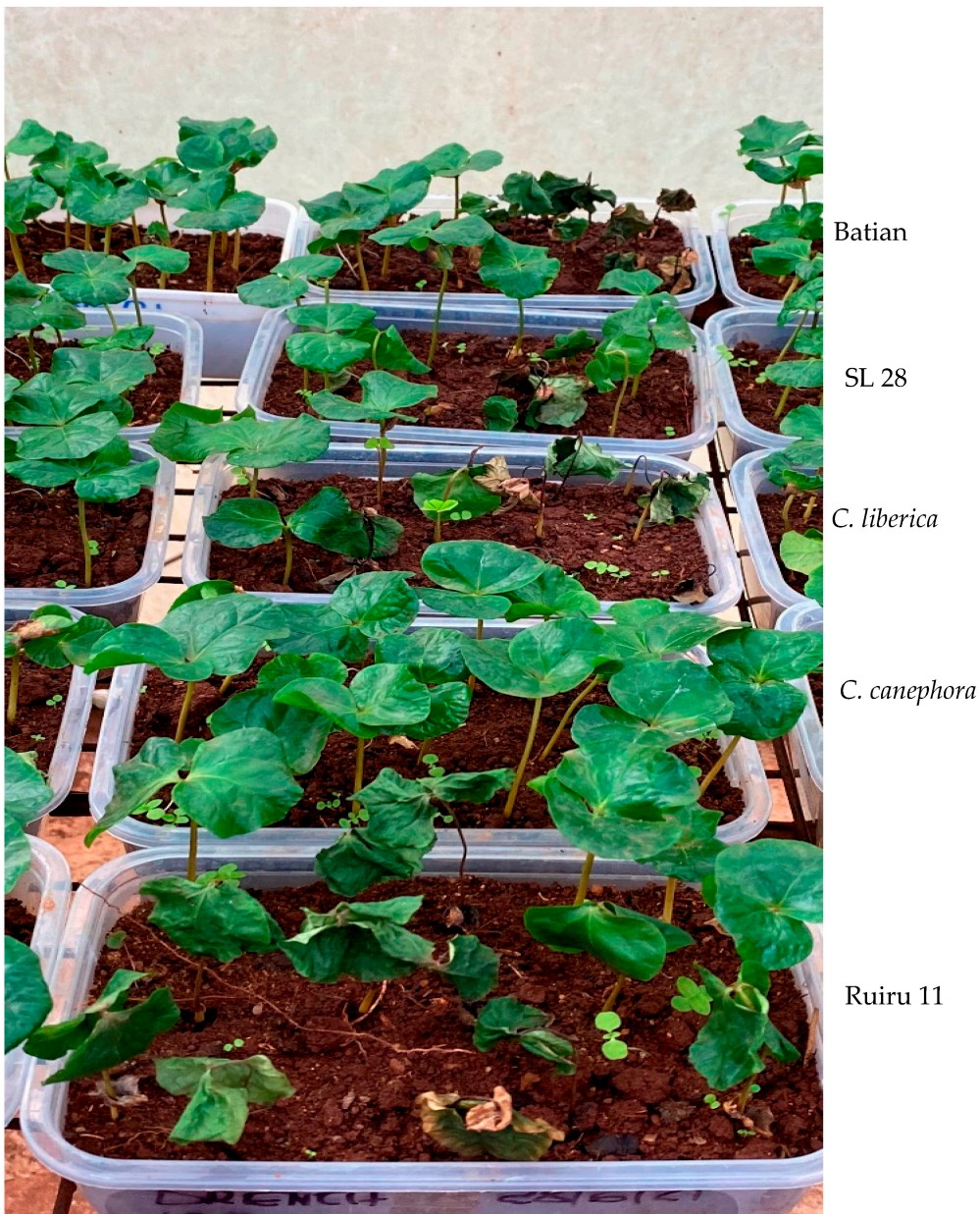

**Figure 2.** Reaction of nine week old coffee seedlings of Batian, SL28, *C. liberica*, *C. canephora* and Ruiru 11 to infection by *F. stilboides* after inoculation in the laboratory.

The disease progression significantly varied among the five varieties. In the first season, the Area Under the Disease Curve (AUDPC) for Ruiru 11 was significantly high ($p < 0.05$) followed by SL 28 while Robusta had the lowest AUDPC. In the second season, coffee varieties Ruiru 11, Batian and Liberica accumulated more disease and recorded significantly higher AUDPC than Robusta coffee varieties (Figure 4). In all the varieties, significant disease progression began from Week 7 toWeek 16 without a decline (Figure 4).

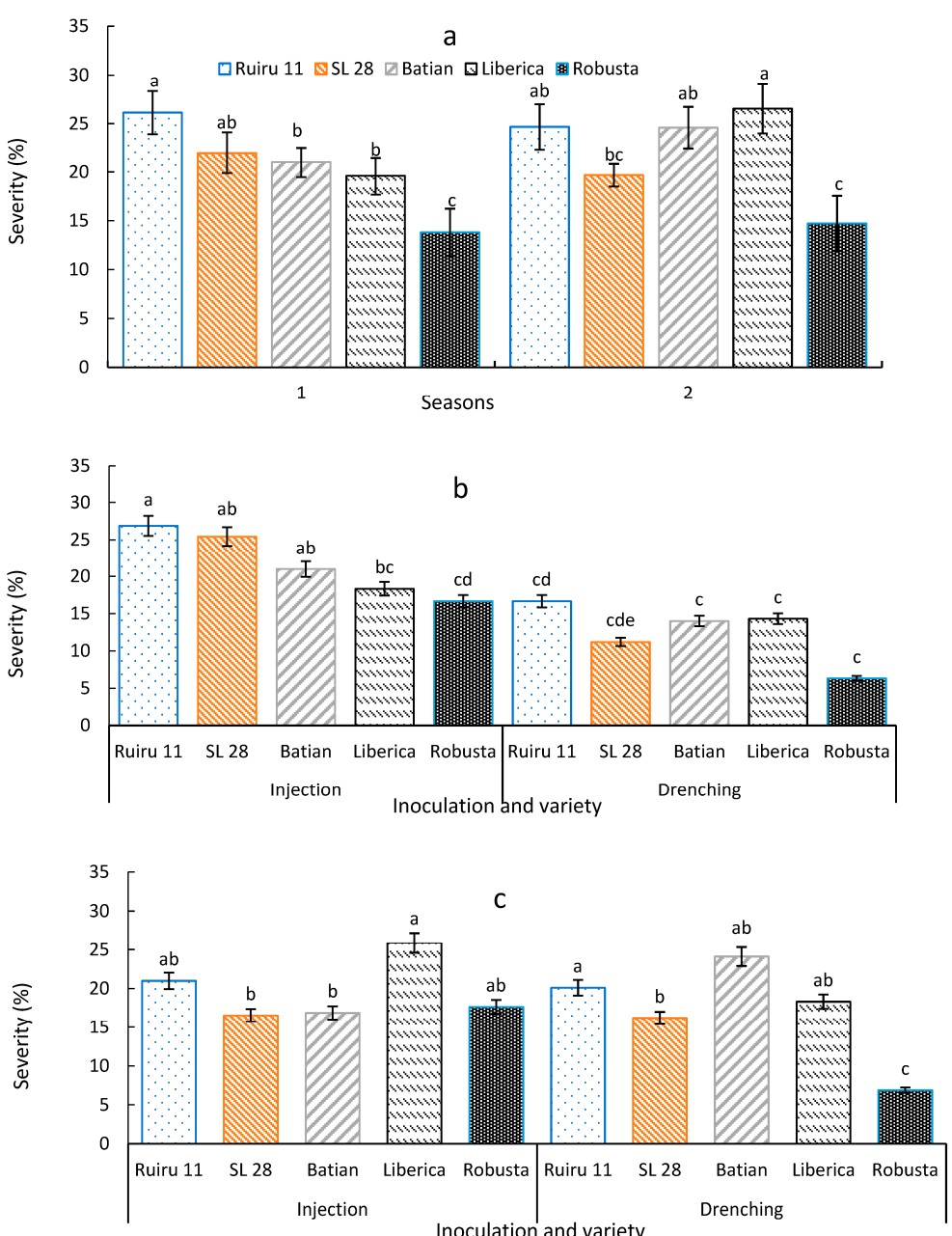

**Figure 3.** Varietal responses to inoculation with *F. stilboides* (**a**) across seasons; to different inoculation methods (**b**) Season 1 and (**c**) Season 2. Bars with the same letters are not significantly different at $p < 0.05$.

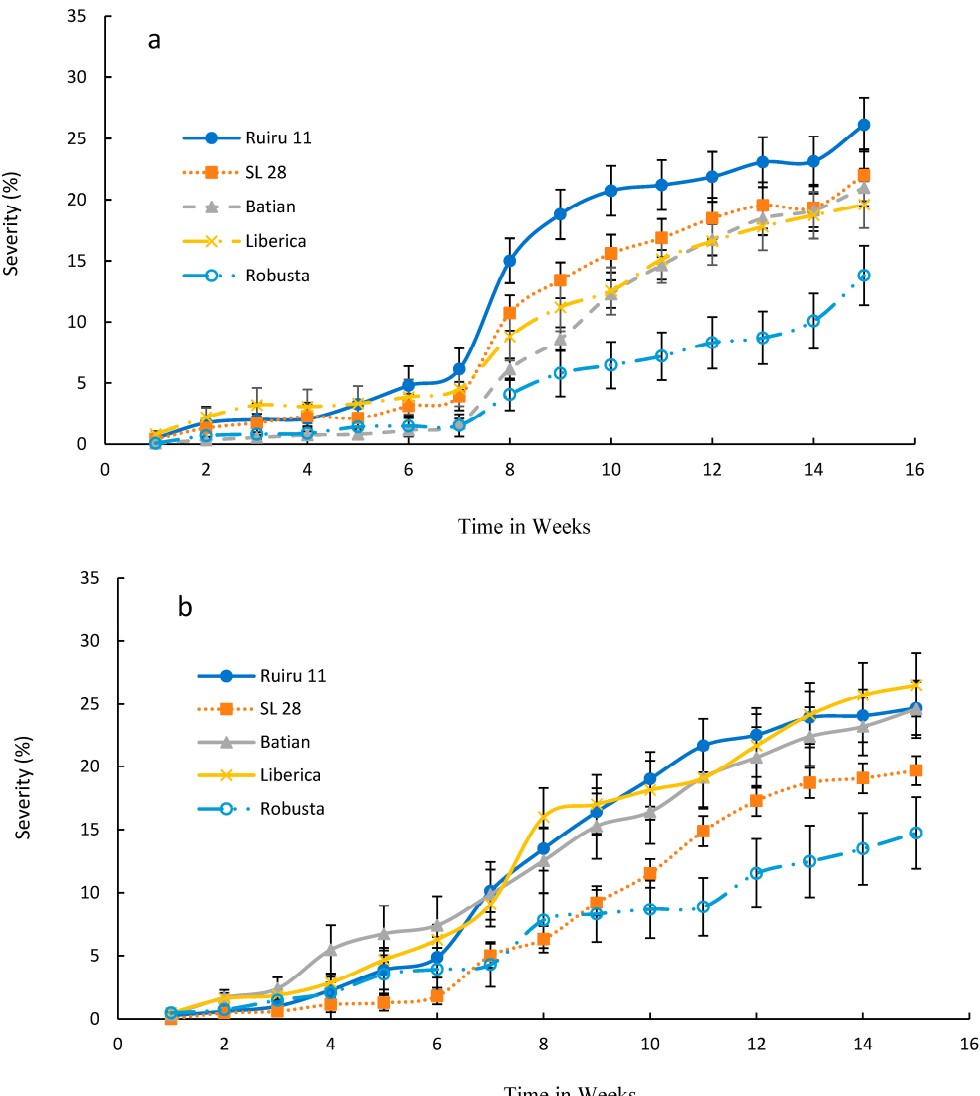

**Figure 4.** The FBD progression on different coffee varieties in different seasons (**a**) Season 1 and (**b**) Season 2.

## 4. Discussion

The pathogenicity studies indicated that the inoculated coffee seedlings developed symptoms like those reported in fields and those documented by Siddiqi and Corbett [6]. According to Siddiqiand Corbett, [6] Fusarium bark disease (FBD) is characterized by lesions on the cambial tissues, while on green stems the lesions appear sunken. Wilting and death can also occur suddenly and may happen in a matter of days. Koch's postulate further confirmed that the test isolates from diseased coffee seedlings were pathogenic to the seedlings and caused similar symptoms as those observed in the field. The rapid development of disease symptoms identical to those observed in the fields on infected coffee seedlings suggested how aggressive the isolates used were. The study also compared the relative aggressiveness of various *F. stilboides* isolates from different locations, and the results showed that isolate TN002B (I) from Kirinyaga county was more aggressive; however, there were no significant differences ($p < 0.05$) between the other two isolates. This outcome suggests that the location from which an isolate is collected may affect a pathogen's aggressiveness. Many authors have reported that pathogen aggressiveness shows variability between isolates' different geographical regions [16,17]. All three isolates were capable of causing significant wilting and death to coffee seedlings. The coffee seedlings used were young, green, and sappy, and symptoms first appeared in some

varieties as early as 10 days, while the first death occurred within the first month. These results are similar to those Siddiqi & Corbett reported [6] and suggest how aggressive the *F. stilboides* isolates were. According to the authors, the more years the coffee tree has accumulated wood, the more difficult it was to be infected, for instance in their work, the first FBD symptoms in woody coffee seedlings were observed 21 days after inoculation.

Among the varieties screened for susceptibility/resistance to FBD; Robusta coffee was found to be more resistant than the other four varieties in both seasons. This result confirms those reported by Storey [18] where comparisons of inoculations of *C. arabica* with *C. liberica*, Robusta and its relative's *C. quillou*, *C. canephora*, *C. congensis* hybrids, indicated a high resistance. Small lesions, which rapidly healed, developed in *C. canephora* and *C. congensis* hybrids. The author reported that inoculation of younger Robusta seedlings resulted in only one death out of four. The expression of resistance by the robusta seedlings, indicates that young coffee seedlings can be used in a preliminary screening of a large population to narrow down to target individuals that can inform coffee breeding programs. The most susceptible cultivars were Ruiru 11, Batian and SL28 which recorded more disease incidence and severity and had the largest area under disease progress curve (AUDPC). The susceptibility of these cultivars could be attributed to their narrow genetic base and with a diversity index of less than 10% [19]. Development of symptoms in older suckers takes longer, even up to four months to appear. Common symptoms in older plants, therefore, include brown sunken lesions with dead extra cambial tissues, wilting of leaves, and even sudden death of coffee trees [6].

Coffee bark disease, also called Storey's bark disease can be highly destructive and can cause serious economic losses due to decline and death of the coffee trees as was noted in 2013 in Nyeri. The disease was regarded as a minor one but suddenly gained major status possibly due to changes in climate, management practicespractices or virulenvce of the pathogens. According to Zhang et al. [20], changes in key climate variables such as temperature and moisture influence disease infectivity and pathogenicity. For instance, studies have shown that temperature significantly impacts growth and reproduction of Fusarium species [20]. According to Fick and Hijmans [21] the temperature levels are increasing and will increase by 2 to 6 °C in coffee-growing regions of Africa by 2080, and this will potentially facilitate the occurrence and spread of Fusarium bark disease. Our data on severity and disease progress illustrates that once FBD has infested a coffee field, it will continue to infect susceptible coffee trees, which may lead to death of the trees. Seasonal variation was observed in the experiments, with significantly lower disease severity in the first season than in the second season. According to Storey [18] FBD is primarily on Arabica coffee and they reported high susceptibility. Therefore, studies are required to ascertain the quantity of loci and specific gene action that is responsible for resistance in Robusta coffee. This can be a source breeding plan aimed at developing resistant coffee varieties to Fusarium bark disease. Studies are also needed to better understand how coffee plants and the *Fusarium stillboides* pathogen interacts in the current and future environmental contexts. This can be performed through long-term data monitoring in order to understand the complex nature of the coffee tree and its interactions with *Fusarium stilboides* This study also compared the usefulness or the abilities of different inoculation assays to evaluate FBD disease development. Some of the variabilities in this study are related to the virulence of the isolates, and the level of resistance in the cultivars. Different isolates were used to evaluate their aggressiveness on coffee seedlings and Isolate 2 (TN002B) in comparison with the other isolates appeared to be the most aggressive. Studies are recommended on the heredity of the aggressiveness among the isolates to determine if it is a stable trait with a genetic component [2,22]. According to Van der Plank [23], the aggressiveness of a pathogen is measured by the quantity of the disease induced on a susceptible host. However, the behavior of an isolate/pathogen under greenhouse conditions may not reflect its behavior in the field. The variations in response to infection by the current commercial varieties provide a key to breeding programs aimed at developing resistant varieties [24–26]



The less susceptible species could serve as sources of genetic material for improving the high-yielding but more susceptible genotypes.

**Author Contributions:** Conceptualization, G.O.A., J.O.O., J.J.M. and E.K.G.; Methodology; G.O.A., J.O.O., J.J.M., E.K.G., D.W.M. and O.O.O.; Software, G.O.A.; Validation, G.O.A., J.O.O., J.J.M., E.K.G., D.W.M. and O.O.O.; Formal Analysis, G.O.A.; Data intepretation G.O.A. and O.O.O.; Investigation, G.O.A. and O.O.O.; Resources G.O.A. and E.K.G.; Data Curation, G.O.A.; Writing–Original Draft Preparation, G.O.A.; Writing–Review & Editing, G.O.A., J.O.O., J.J.M., E.K.G., D.W.M. and O.O.O.; Visualization, G.O.A., O.O.O. and E.K.G.; Supervision, J.O.O., J.J.M., E.K.G. and D.W.M. All authors have read and agreed to the published version of the manuscript.

**Funding:** The research was funded by the National Research Fund, Kenya grant number (2016/17) and the World Bank Group through the Centre for Excellence in Sustainable Agriculture and Agribusiness (CESAAM), Egerton University, Kenya grant number (WB-IDA Credit 5798-KE). The APC was funded by the fungicides project, KALRO-Coffee Research Institute, Ruiru, Kenya.

**Data Availability Statement:** The data presented in this study are available on request from the first author. The data are not publicly available due to (It is part of a student's ongoing dissertation research).

**Acknowledgments:** The authors thank the National Research Fund, Kenya, and the World Bank Group through CESAAM, Egerton University for funding the project. The authors would also like to acknowledge the Plant Pathology Section staff at KALRO-Coffee Research Institute for their technical and administrative support. This paper is published with the permission of the Director General, KALRO.

**Conflicts of Interest:** The authors declares no conflict of interest.

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
