# Peer review of "A Method for Selection of Coffee Varieties Resistant to Fusarium stilboides"

_agronomy, doi:10.3390/agronomy13092321_

Round 1

Reviewer 1 Report

Pay attention to sections as indicated on attached manuscript

English sufficient

Author Response

Reviewer One

Editorial Comments in Abstract, introduction, and Materials and methods

Are you sure this is six weeks and not 6 days. Usually an actively growing culture is used for preparation of inoculum. A six week old culture are not actively growing anymore

Plates or pieces? 15 plates in 150ml of water will be too much?

The reviewer’s comments on the manuscripts have been implemented and are highlighted in the manuscript

This has been corrected to 6 days

Corrected to reflect 15 cut pieces of F. stilboides cultures in 150ml of SDW

Reviewer 2 Report

Major comments

1.      There were not any references cited in the materials and methods part. Are these all originally created by the authors?

2.      What are the phylogenetic relations of Ruiru 11, Batian and SL 28? Do they have relations which the HDT series?

3.      Where is the result in the first part of the Results? Or it should be merged with the second part of Results?

4.      What is the coffee variety used in the second part of Results?

5.      The Results part is poorly written, which should be rephrased. Especially the Figure 4 was not described in the part.

6.      Seedlings were used to evaluate disease resistance, which have weak resistance compared with stems or branches of coffee tree. So how did the pathogen affect the coffee tree? Which should be described and discussed in the manuscript.

Minor concerns

1.      The figure in page 9 should be Figure 2.

2.      The bars in Figure 3 should be in the same style.

A grammar checking is necessary.

Author Response

Reviewer Two

References in Materials and methods

References in the M&M sections have been included and are highlighted,

What are the phylogenetic relations of Ruiru 11, Batian and SL 28? Do they have relations which the HDT series?

The phylogenetic relationship between the coffee varieties have been indicated and clearly stated as being narrow genetic base and with a diversity index of less than 10%.

Where is the result in the first part of the Results?

The first part of the result is describing results in Table 3.

What is the coffee variety used in the second part of Results? The Results part is poorly written, which should be rephrased

All the coffee varieties in the study were compared in this section. The results section has been rephrased and figure 4 included

Seedlings were used to evaluate disease resistance, which have weak resistance compared with stems or branches of coffee tree. So how did the pathogen affect the coffee tree? Which should be described and discussed in the manuscript?

Despite the potential of weak resistance we used the coffee seedlings to ensure quick assessment of the disease response. Again even though young coffee seedlings can exhibit weaker resistance they can still show level of resistance and this can guide in breeding efforts

Reviewer 3 Report

The discussion is rather modest, as are the references, which contain only 12 items. It would be good to refer to observed climate change and try to predict how it will affect coffee cultivation and how it will affect Fusarium pathogens? Why did the disease become important in Kenya (Nyeri County) only after 2013? What could have influenced it - better growing conditions for the fungus or weakened plants?
Will the selected coffee varieties that are resistant to the pathogen Fusarium stilboides also be resistant to changing environmental conditions, e.g. more frequent and longer droughts?
Only if these interactions between environment, plants and pathogens are known can appropriate breeding plans be drawn up. This is of great importance not only for coffee producers, but also for coffee consumers.

English is fine

Author Response

Reviewer three

The discussion is rather modest, as are the references, which contain only 12 items. It would be good to refer to observed climate change and try to predict how it will affect coffee cultivation and how it will affect Fusarium pathogens? Why did the disease become important in Kenya (Nyeri County) only after 2013? What could have influenced it - better growing conditions for the fungus or weakened plants?

The occurrence of Fusarium Bark disease was influenced by climate change. We have tried to relate climate and the occurrence of the disease and also indicated how the changes in weather parameters especially temperature  can have effect on occurrence and spread of Fusarium pathogen

Will the selected coffee varieties that are resistant to the pathogen Fusarium stilboides also be resistant to changing environmental conditions, e.g. more frequent and longer droughts?

We have suggested that studies are needed to look at how coffee plants and Fusarium stillboides pathogen interacts in the current and future environmental contexts which can be done through long-term data monitoring

Only if these interactions between environment, plants and pathogens are known can appropriate breeding plans be drawn up. This is of great importance not only for coffee producers, but also for coffee consumers.

We have suggested long term studies on the interaction between environment, pathogens and coffee are needed to better come up with a breeding plan aimed at developing resistant varieties

Round 2

Reviewer 2 Report

The results part is still poor, which should be fully described. It is hard to believe that three short paragraphs could contain all fingings.

Figure 3 is also not modified. At least the bar width should be same.

A grammar checking is necessary.

Author Response

Major areas reviewed 

  1. The results section has been reviewed, and additional interpretation done
  2. Figure 3 has been formatted to ensure all bars are equal
  3. Comprehensive English language review has been done using Grammarly (Premium subscription)
  4. Additional information and references on possible causes of change in the disease severity in Kenya have been added.
  5. Substitute reference No 2
  6. Deleted references 9, 10, 11

Round 3

Reviewer 2 Report

The authors have addressed most of the comments. The manuscript could be accepted at the current form.

Fine.